

# Hand gesture recognition via deep data optimization and 3D reconstruction

Zaid Mustafa[1], Heba Nsour[2] and Sheikh Badar ud din Tahir[3,4,5]

[1] Department of Computer Information Systems, Prince Abdullah Bin Ghazi Faculty of Information and Communication Technology, Al-Balqa Applied University, Al-Salt, Al-Balqa, Jordan
[2] Department of Computer Science, Prince Abdullah Bin Ghazi Faculty of Information and Communication Technology, Al-Balqa Applied University, Salt, Al-Balqa, Jordan
[3] Department of Computer Science, Air University, Air University, Islamabad, Pakistan
[4] Department of Software Engineering, Capital University of Science and Technology (CUST), Islamabad, Pakistan
[5] Current Affiliation: Department of Computer Science, SZABIST University, Islamabad, Pakistan

## ABSTRACT

Hand gesture recognition (HGR) are the most significant tasks for communicating with the real-world environment. Recently, gesture recognition has been extensively utilized in diverse domains, including but not limited to virtual reality, augmented reality, health diagnosis, and robot interaction. On the other hand, accurate techniques typically utilize various modalities generated from RGB input sequences, such as optical flow which acquires the motion data in the images and videos. However, this approach impacts real-time performance due to its demand of substantial computational resources. This study aims to introduce a robust and effective approach to hand gesture recognition. We utilize two publicly available benchmark datasets. Initially, we performed preprocessing steps, including denoising, foreground extraction, and hand detection via associated component techniques. Next, hand segmentation is done to detect landmarks. Further, we utilized three multi-fused features, including geometric features, 3D point modeling and reconstruction, and angular point features. Finally, grey wolf optimization served useful features of artificial neural networks for hand gesture recognition. The experimental results have shown that the proposed HGR achieved significant recognition of 89.92% and 89.76% over IPN hand and Jester datasets, respectively.

## INTRODUCTION

Regarding human–computer contact, technological breakthroughs in artificial intelligence and modern technology have created several efficient communication channels. Hand gesture recognition (HGR) is a technique that involves the receiver recognizing physical motions produced by the recipient's fingers, hands, arms, head, and face. Meanwhile In recent times, numerous disciplines, including modeling, computing, biomedicine, and gadgets, have increased their focus on realistic human interactions and understanding in the innovative city environment (*Gholami & Noori, 2022*). Gestures are the most intuitive

Corresponding authors
Zaid Mustafa, z_lami@bau.edu.jo
Sheikh Badar ud din Tahir,
sheikh.tahir@students.au.edu.pk

approach to managing smart home gadgets. The daily-used appliances are an integral component of your residence (*Tan et al., 2020*). Modern consumers are highly concerned with their well-being and safety and are increasingly interested in household sensors. Users will command the illumination, microphones, air conditioning systems, and other similar robots with hand movements to improve their everyday routines. Anyone can operate all the electronics in their home with a gesture. A motion detector represents one of essential components of a sensor network. HGR has numerous applications, including communications with and amongst deaf individuals and connections among early childhood and PC-using clients (*Pinto et al., 2019*). Healthcare providers offer hand muscle exercises in which HGR plays a critical part in treatment goals. According to the World Health Organization (WHO), more than 15 million individuals are impacted by brain hemorrhage and 50,000 by spinal cord injuries (*Bajunaid et al., 2020*). Injuries impair the movement of the upper limbs and result in long-term impairments. Treatment is a crucial component of upper extremity healing. HGR is used to make rehabilitative gestures and identify everyday movements (*Li et al., 2017*).

Ineffective communication, gestures are classified as static and dynamic. A stable gesture is observed in one instant, but an emotional gesture varies over time. Static movements are distinct periods of transformation within a moving movement that manifest as a particular movement or gesture. Perception technology and vital statis-tics can infer the activity using I-image, (ii) monitors, and (iii) fingers (*Wadhawan & Kumar, 2021*). The sensor systems and sleeves determine in real-time complex and thumb locations. However, the employment of gloves and detectors imposes unavoidable stress on the consumer, and the thickness of wires might impede hand mobility, which impacts the effective-ness of motion measurement. On the other hand, photos of an individual's hand movements can be captured using one or more devices. The camera gathers static exercises, which are utilized for training the system for recognition; just a significant dataset is required to accomplish this (*Dang et al., 2020*).

*Li, Chen & Wang (2021)* suggested a hybrid model incorporating 3D convolutional neural networks (CNNs) with recurrent neural networks (RNNs) to recognize hand gestures. They tested the model on two benchmark datasets and got cutting-edge results. To distinguish hand motions from depth maps, *Mustafa & Brodic (2021)* used 3D CNNs. They attained an accuracy of over 97% by using a proprietary dataset of hand gestures to train their model.

Using 3D CNNs and depth maps, *Grigorov, Zhechev & Mihaylov (2021)* created a real-time hand gesture recognition system. On a custom dataset, they tested their system and got an accuracy of over 90%. For the recognition of 3D hand gestures, *Lu, Cheng & Zhang (2021)* proposed an enhanced deep learning system. On a benchmark dataset, they tested their model and obtained a recognition rate above 96%. *Baraldi, Grana & Cucchiara (2021)* recognized hand motions from depth maps using 3D CNNs and transfer learning. They obtained cutting-edge results by evaluating their model on three benchmark datasets.

The various models offered for regulating smart home equipment *via* hand gesture detection can be split into two major categories. The first strategy is based on the detection of hand movements utilizing multiple motion sensors in intelligent household

items (*Oyedotun & Khashman, 2017*). A single inertial sensor is utilized in motion-based monitoring. These detectors are responsible for monitoring the hand's impulse, velocity, and position. The detection limit of body movement cameras in household appliances is a downside of employing these capabilities to regulate electronic appliances such as televisions, radios, and interior illumination (*Gholami & Khashe, 2022*). The method employs sensing de-vices or camcorders to procure directions from hand movements; the sensors and cameras are prepared on segmentation methods, including color, structure, appearance, placement shapes, and hand motion. Following the second method, our suggested model recognizes hand motions using sensing devices or webcams (*Trong, Bui & Pham, 2019*).

This research study involves a robust method for hand gesture detection and recognition. We consider two state-of-the-art datasets for proposed HGR method evaluation. Initially, we perform some steps for data normalization and other related tasks, such as noise reduction and frame conversions. Hand shape detection is the second step of our proposed model. Next, we extract useful information in terms of a features extraction model, 3D reconstruction is applied to get more accurate values and accuracy rate, data optimization is performed *via* the heuristic algorithm, namely grey wolf optimization, and finally, recognition accuracy and classification, we apply artificial neural network. The research study's objectives are presented in the following points:

- The study aims to propose the development of an optimized communication channel for human–computer interaction by utilizing hand gesture recognition (HGR) system.
- We compare and analyze the characteristics of static and dynamic gestures in relation to their effectiveness in communication and recognition.
- We investigates various methodologies employed in the acquisition and analysis of hand gestures, encompassing the utilization of image sensors, monitors, and finger-based systems.
- The study presents a comprehensive approach for extracting robust features in order to improve gesture recognition. The proposed method incorporates geometric, 3D points, and angular features.
- We adopted 3D modeling techniques to enhance the precision and accuracy of hand gesture information.

The present research article is structured into several sections and comprehensive coverage of the research. 'Related Work' provides a thorough exposition of the related method, while 'Materials & Approach' outlines the proposed method in detail, involving pre-processing, hand detection, data mining and classification methods. 'Experimental Results & Analysis' delivers the experimental part, including details of the experiments, results and evaluation with other state-of-the-art methods. Finally, 'Conclusion' presents the conclusion and provides some potential future insights.

## RELATED WORK

Due to their decreasing rate and actual size, IMUs have subsequently become a standard technology found in telephones, smart watches, and smart bands (*Chung et al., 2019*). The

science community is becoming more interested in using IMUs for higher physical levels due to the adoption of sophisticated and wearable technology. A stretchable variety of electronics for image stabilization was suggested by *Chen et al. (2018)*. They developed a method that integrated an IMU onto a rubber stopper that could be fastened to the body. The molded case served as stress reduction, protecting the sensors.

Additionally, the band is simple to connect and release to get information from other locations, such as the firearms or legs. Furthermore, since the motion causes fluctuations in the rubber stopper, the detected signals were impacted by technique encouragement (*i.e.,* noise). Due to the intensity and velocity associated with the movement, hand gestures are particularly prone to image noise. The detected motion artefact may be reduced if the IMU is immediately linked to the surface. However, creating an easily correctable sensor is generally expensive and time-consuming. An affordable, easily repairable six-axis IMU can be made using our methodology. Due to its promise in medical and interpersonal behavior fields, hand gesture detection by DL is a topic of active research (HCI). For example, *Cole et al. (2020)* developed an artificial neural net-work-based technique to distinguish cigarette motions from an apple watch's tri-axis magnetometer. The use of lenses to recognize static motions has been extensively re-searched. Different techniques extract motion detection information for motionless hands (*Aldabbagh et al., 2020*). The entire hand or only the digits can be used to feature extraction.

HGR of four indicators is a difficult task because the complete hand's proposed technique is heterogeneous and necessitates substantial pattern recognition for authentication. Numerous academics have put forth various approaches for recognizing gestures made with the entire hand. A method was put out by *Cheng et al. (2016)* that retrieved the hand's shape and used the center to determine compactness and finger location for gestures. Using a prediction model, nine different actions are classified as movements. Using Hu invariant periods along with skin color, angle, and other factors (*Oprisescu, Rasche & Su, 2012*) identified the hand. The researchers employed a distance measure configuration method for categorization. In their system, *Yun, Lifeng & Shujun (2012)* split the hand during pretreatment. A localized shape pattern and block-based characteristics are extracted for a stronger depiction of the hand. These features are integrated to identify static hand motions, and a classification method is employed. Using YCbCr values, Candrasari (*Agarwal, Raman & Mittal, 2015*) placed the hand. They used the discrete wavelet transform (DWT) to feature extraction and then successfully classified the data using the hidden Markov model (HMM) and k-nearest neighbor (KNN).

*Jalal, Khalid & Kim (2020)* used a user-worn glove to retrieve the hand, utilizing contour modelling. American Sign Language (ASL) and the numerals 0 through 9 were classified using ANN. *Chen, Shi & Liu (2016)* used a color model to partition the hands and collected training hand positions. The approach suggested by *Bhavana, Surya Mouli & Lakshmi Lokesh (2018)* was divided into the following phases: preprocessing, hand segmentation using cross-correlation method for detect-ing edges feature vector computation using the Euclidean distance across contours, and finalization. Following that, a comparison between the Hamming distance and the spatial relationship is made to recognize gestures. A unique approach to hand gesture identification was put out by (*Yusnita et al., 2017*), whereby the

shoulder is subtracted employing location modification, and the hand is identified using skin-color features. Calculated gesture moments are used with SVM to classify the motions. A technique to re-cover the hand using wristband-based contour characteristics and arrive at identifiable information, a straightforward feature-matching strategy was suggested. A functional-ity structure was suggested by *Liu & Kehtarnavaz (2016)* for assessing 3D hand posture. For feature extraction, they utilized convolutional layers, which were strengthened by a new long short-term dependence aware (LSTD) module that could recognize the correlation on various hand parts. The authors also incorporated a contextual integrity gate to in-crease the trustworthiness of features representations of each hand part (CCG). To compare their technology to other cutting-edge techniques, they employed evaluation metrics.

The localization of hand landmarks to extract features enabling gesture detection has been approached from many angles. Hand gesture detection is used extensively by investigators in the preponderance of currently used techniques. A technique for extracting significant hand landmarks from images was suggested by *Ahmed, Chanda & Mitra (2016)*. They pinpointed the exact coordinates among those landmarks and then correlated those landmarks with their respective counterparts in 3D data to simulate the hand position. Regions of interest (ROI) can be generated using a method developed by *Al-Shamayleh et al. (2018)* using the local neighbor methodology. To identify fingertips, researchers ap-plied active contour detection techniques. *Pansare & Ingle (2016)* created an innovative method to determine the fingernails and the palm's center. An adjustable hill-climbing approach is used on proximity networks to execute the fingertip-detecting process. The proportional lengths across fingers and valley locations are used to identify individual fingers.

## MATERIALS & APPROACH

The proposed method is based on robust approaches for hand gesture detection and recognition. We consider two complex databases for our proposed method evaluation. Initially, we perform various prerequisite steps for data normalization and other related tasks, such as noise reduction and frame conversions. Hand shape detection is the second step of our proposed model. Next, we extract useful information in terms of a features extraction model, 3D reconstruction is applied to get more accurate values and accuracy rate, data optimization is performed *via* the heuristic algorithm, namely grey wolf optimization, and finally, recognition accuracy and classification, we comprehensive method is presented in Fig. 1.

### Pre-processing

In this subsection, we cover the pre-processing for the suggested technique, which begins with foreground extraction using change recognition and associated component-based approaches and strategies. The connects a component labelling approach for fragmenting the human hand silhouette and finding conspicuous skin pixels.

We adopt these techniques from different studies. For example, using Otsu's method and image segmentation, *Petro & Pereira (2021)* proposed a novel method for optimal color quantization. On numerous test images, author revealed that their method performs

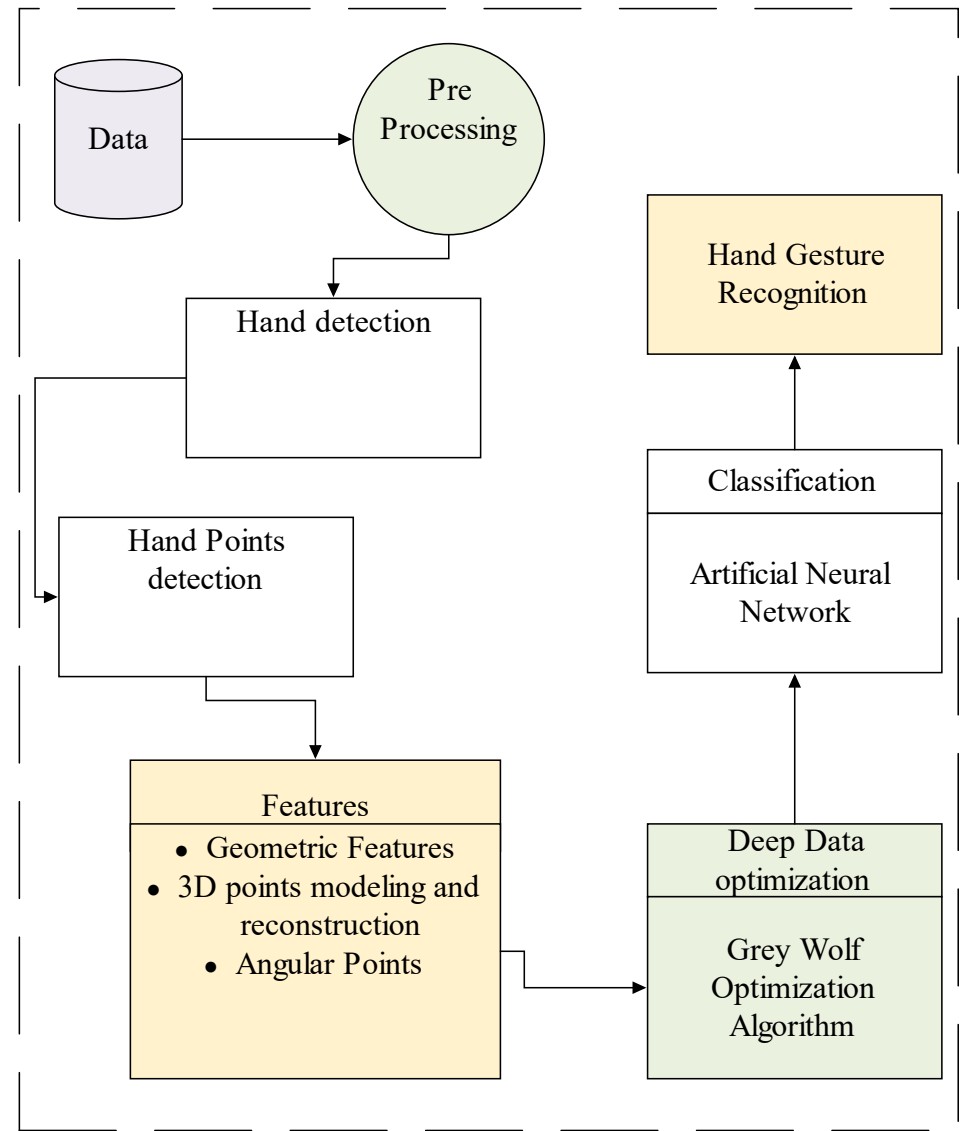

**Figure 1  The overall flow of the proposed system.**

better than the other approaches. *Lisowska, Tabor & Ogiela (2021)* provided a novel image thresholding technique that combines fuzzy c-means clustering and multi-Otsu's method. The proposed techniques were tested on several benchmark datasets which shows better results. In another study, *Zhang & Jia (2021)* included a new version of Otsu's method among their list of novel histogram-oriented thresholding techniques for remote sensing images.

Utilizing histogram-oriented thresholding, we differentiated the hand shape after extracting the skin elements. Using Otsu's method, numerous threshold standards of Δ (Eq. 1) were adapted, and the extreme color strength of stochastic histogram hso (x,y) is

described as:

$$hiso(x,y) = \begin{cases} IT\left(\dfrac{TOi + Tehi_{max}}{4}\right), & if\, Tehai_{max} \leq 5 \\ IT\left(\dfrac{TOi + Tehi_{max}}{2}\right), & if\, Tehai_{max} > 5 \end{cases} \qquad (1)$$

where *IR* is the overweight, *TOai* is a threshold which is suggested by Otsu's technique, and $Tehai_{max}$ is the main position of skin occurrence over extracted histogram directory . This process is practical for every grey scale subdivision of given image, which articulated as;

$$Ige(x,y) = \begin{cases} (0,0,0), & if\, gr(x,y) = 0 \\ (Igr(x,y), Imgr(x,y), Ib), & if\, g(x,y) = 1 \end{cases}. \qquad (2)$$

## Hand detection

The humanoid hand silhouette ridge identification process consists of two steps (*Zhang et al., 2018*): sequential edge identification and ridge data synthesis. In the binary border separation process, binary boundaries are recovered from the RGB silhouettes produced in the above-mentioned denoising phase. Employing distance transformation, the proximity maps are generated on the boundaries. On the other hand, the ridgeline data creation phase, the local optima are acquired from the pre-estimated mapping to generate ridge data along the binary vertices. The mathematical representation of hand recognition is

$$\gamma K = n \sum tx = 1||\alpha x| - |\beta|| \, where\, K = 1,2,3\ldots \qquad (3)$$

where $\alpha$ symbolizes the centroid point of the trajectories deposited in the confusion table, $\beta$ denotes the updated trajectories of the evaluated data, $\gamma$ indicates the detachment among the present standards of the confusion matrix and the novel trajectories.

## Hand points detection

The segmentation hand is then utilized to detect landmarks. Numerous methods are offered for localizing hand landmarks, which aid in extracting the features for recognizing and identifying individual movements. The bulk of strategies are quite straightforward and restrict the precise location of landmarks. After collecting the acoustic waves of geodesic velocity using the fast-marching algorithm (FMA) on frames, landmark recognition is conducted. The color values of quads p are generated based on the outlines' outer boundaries b. Pixels with identical color values c are identified and then the mean is calculated; based on the pixel's average values, the landmark l is painted. For the innermost landmark, the value of bright green is determined and the distances adjacent points is determined. Calculating the fingers yields

$$l = c(px, py)/2 \qquad (4)$$

where *px* and *py* have the same color in the external surface and *cpx, py* is the total amount of pixels with that colour in the external surface. On the hand outlines in Fig. 2 below, landmark is inspired:

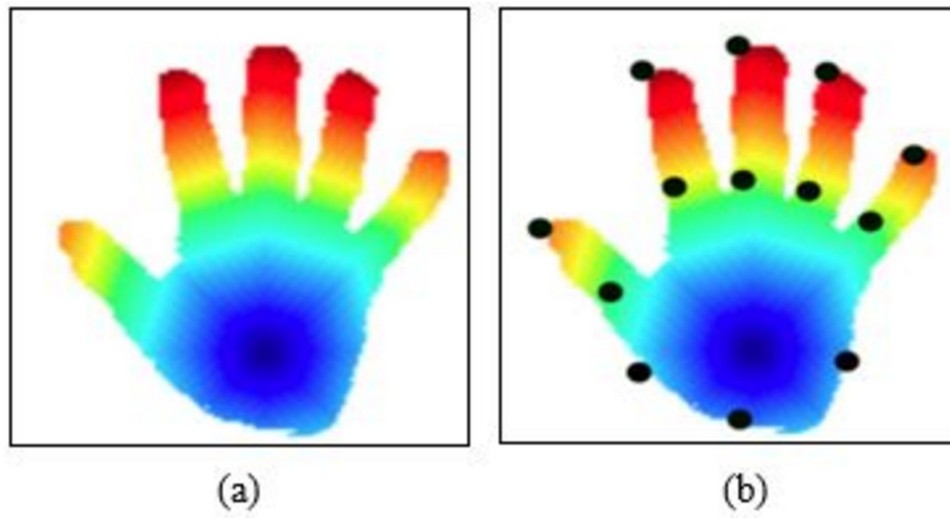

**Figure 2  Sample results of hand point extraction, (A) fast marching algorithm results, (B) key points.**

## Features extraction

This step presents the details of features abstraction techniques for hand gesture recognition over challenging databases. We employed three detail methods for acquisition of features: geometric features, 3D points modeling and reconstruction, angular point features. Algorithm 1 provides the complete methodology for features abstraction.

Algorithm 1 **Features Abstraction**
Input: Raw_data
Output: Feature_vect ($fe_1, fe_2, \ldots \ldots fe_n$)
$Extarcted_{features} Vector \leftarrow []$
$Data \leftarrow$ AcqData_F_F()
Data_dim_F1 $\leftarrow$ Acq_F1_dim()
Step PAP(Video, RGB)
$Features\_Vectset \leftarrow []$
Denoise_Input_Data $\leftarrow$ Denoising()
Sampled_Data(FilteredData)
While exit invalid state do
$[Gmf, 3DpM, Apf] \leftarrow$ ExtractlFeatures(sample data)
$Feature\_Vect \leftarrow [Gmf, 3DpM, Apf]$
Return MainfeaturesVector

### Geometric features

The point-based approach is used to retrieve hand sign features, which comprise locations on the thumb, forefinger, ring finger, ring thumb, or little finger (see Fig. 3). All the values are merged in numerous ways to provide a range of learning and recognition-related properties. These markers are positional, geometrical, and angle-based properties. The

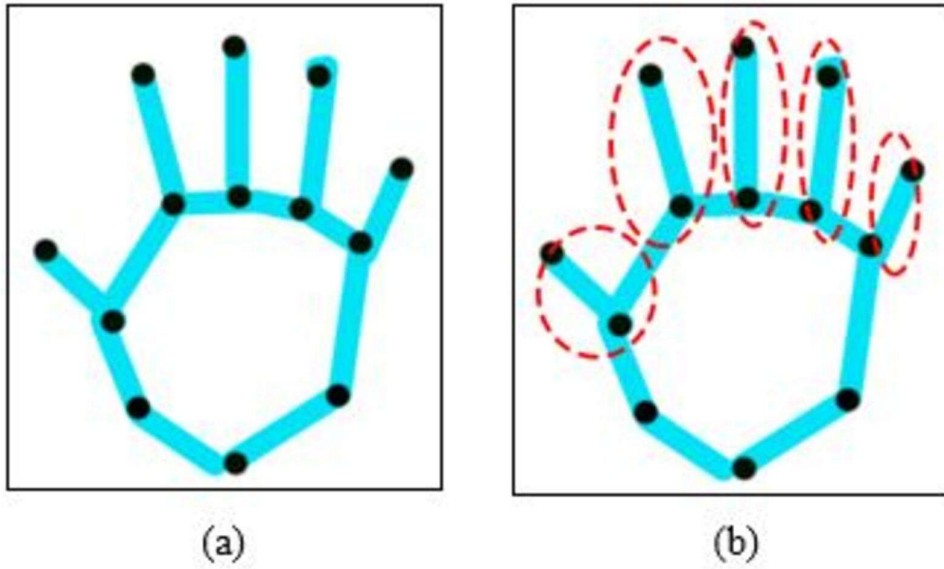

**Figure 3** The results of geometric features over extracted hand points values, (A) extracted hand points and (B) over view of geometric features.

proximity attribute d calculates the distance between ixy radical landmark on the fingertips and the cxy interior marker using hand's geodesic value, which is expressed as

$$\|d\| = \sqrt{(xj_2 - xb_1)^2 + (yj_2 - yb_1)^2} \tag{5}$$

while $d$ is indeed the distance between both places; $xi$ and $xc$ are the $x$ coordinates of the hand's outer reference and inner historic site, congruently; and $yc$ and $yi$ are the $y$ organizes of the identical structures.

### 3D point modeling and reconstruction

The first stage in this section of the three-dimensional hand reconstructions is accomplished using a mathematical model and ellipsoid approaches. Using the details from the connections, we can see that an ellipsoid connects the hand point to the following point and the index point to the inner elements. The rest of the human hand points are connected to the inner hand in a similar manner to how the thumb position is attached to it using an ellipsoid structure. The Eq. (6) shows the formulation of three-dimensional hand reconstructions and computational model.

$$k_{me} = l_a(e_x, e_y) \blacksquare (i_x + 1)(e_x, e_y) \tag{6}$$

Where $k_{me}$ denotes the computational model and $l_a(e_x, e_y)$ is the first points and x,y are the values. Figure 4 shows the detail overview of computational model and 3D hand reconstructions.

### Angular point features

The angular point descriptors are based upon the angular geometric of human hand points. We consider all the extracted points and find the angular relationship between

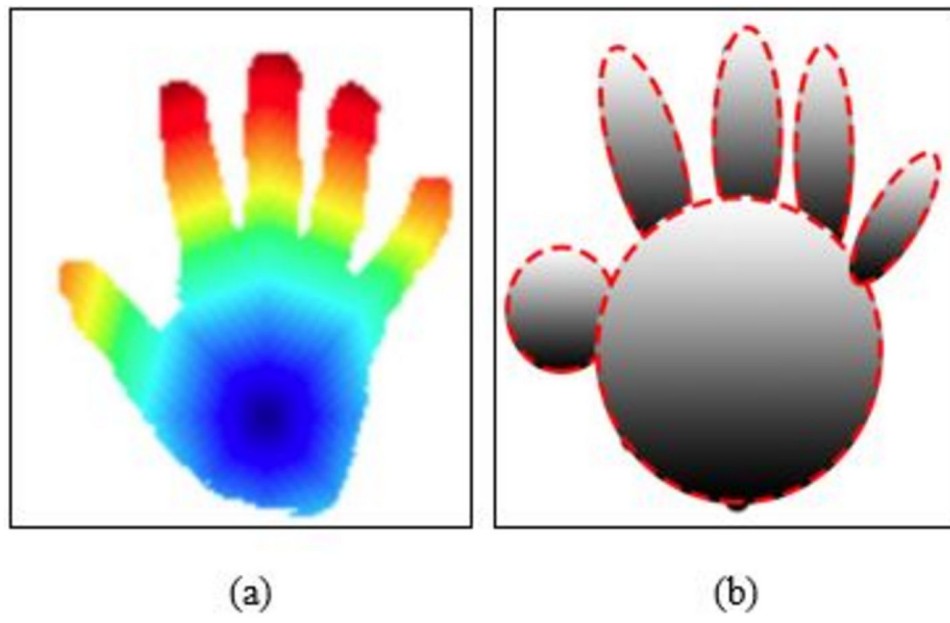

(a)         (b)

**Figure 4 Example results of 3D point modeling and reconstruction (A) fast marching result and (B) 3D reconstruction of hand shape.**

them. Equations. (7)–(9) shows the formulation of angular point features.

$$i = \cos^{=1}b^2 + c^2 - a^2/2bc \tag{7}$$

$$j = \cos^{=1}b^2 + c^2 - a^2/2bc \tag{8}$$

$$k = \cos^{=1}b^2 + c^2 - a^2/2bc \tag{9}$$

Where $i, j$ and $k$ are the procedures of the angle amid two together edges $b <->c, a <->c$, and $a <->b$ of the triangle shaped, correspondingly. After this we map the results with the main features vector.

## Data optimization: Grey Wolf Optimization (GWO)

The GWO algorithm is an intelligent swarm technique created by *Rezaei, Bozorg-Haddad & Chu (2018)*, which imitates the wolf's governing system for cooperative exploration. The grey wolf is a member of the Canidae family and enjoys living in packs. Wolves have a strong hierarchy, with a male or female alpha as their commander. The alpha is mainly tasked with making decisions. The package must accept the leader's instructions. Betas are senior wolves who assist the leader in making decisions. The beta serves as alpha's consultant and administrator. Omega, the lowest-ranking grey wolf, must notify the majority of other dominating wolves. A wolf is a delta if it's neither an alpha, beta, or omega. The omega is governed by the delta, which interfaces with the alpha and beta. Wolves' hunt strategies and

social stratification are represented statistically to develop GWO and achieve improvement. The GWO methodology is evaluated using standard test methodologies, which reveal that it is comparable to other swarm-based approaches in terms of identification and application.

When prey is instigated, the reappearance rises ($t = 1$). Subsequently, the alpha, beta, and delta wolves would supervise the omegas to search and eventually squeeze the prey. Three measures $\bar{A}$, $\acute{Z}$, F and $\hat{Y}$ are predictable to define the surrounding performance:

$$
\begin{aligned}
\hat{Y}\alpha &= \left| \acute{Z}1.\bar{A}\alpha - \bar{A}(t) \right|, \\
\hat{Y}\beta &= \left| \acute{Z}2.\bar{A}\beta - \bar{A}(t) \right|, \\
\hat{Y}\delta &= \left| Z3.\bar{A}\delta - \bar{A}(t), \right|
\end{aligned}
\tag{10}
$$

where $t$ requires the existing repetition, $\bar{A}$ is the position trajectory of the grey wolf, and $\bar{A}1$, $\bar{A}2$ and $\bar{A}$ are the position trajectories of the alpha, beta, and delta wolves. Algorithm 02 shows a comprehensive indication of the data optimization technique *via* grey wolf optimization.

---

Algorithm 2 **Grey Wolf Optimizer (GWO)**
Adjust the grey wolf population $Y_i, i = \overline{1, n}$
Regulate $a, A$ and $C$
Approximation, the fitness of correspondingly search agent (SA)
$Y_\alpha =$ the optimized SA
$Y_\beta =$ the supplementary superlative SA
$Y_\delta =$ the 3rd superlative search agent
While $t < max\ number\ of\ iteration$ **do**
    for *each search agent* **do**
        Arbitrarily initialize $r_1$ and $r_2$
        Adjust the location of the existing SA by the (7)
    Update $a, A$ and $C$
    Approximation the fitness of all SA
    Adjust $X_\alpha$ $X_\beta$ and $X_\delta$
    $t++$
    return $Y_\alpha$

---

## Classification: ANN

This section involves the classification method *via* ANN. An ANN is a set of numerous perceptron or neuron on every stratum; when required information is characterized in the forward channel, this is referred to as a feed-forward neural network (*Abdolrasol et al., 2021*).

Artificial neural networks (ANNs) have the capability to recognize hand gestures and can be trained to solve intricate problems that traditional computing systems or individuals typically encounter. Superintended training methods are frequently employed in practice,

although there are instances where unsupervised training techniques or direct design methods are also utilized. As discussed in the literature, an artificial neural network has been utilized to detect gestures (*Nguyen, Huynh & Meunier, 2013*). The segmentation of images in this system was carried out by utilizing skin color as a basis. The features chosen for the artificial neural network (ANN) comprised adjustments in pixel values across cross-sectional planes, boundary characteristics, and scalar attributes such as aspect ratio and edge ratio.

In addition, the ANN method is effective for handling problems involving RGB data, textual information, and contingency table. The benefit of ANN is its ability to deal with transfer function and its ability to understand variables that translate any inputs to any result for any data. The artificial neurons endow their ANN with substantial qualities that enable the network to understand any complicated relation between output and input data, often known as a universal approximation. Numerous academics use ANNs to tackle intricate relationships, such as the cohabitation of mobile and WiFi connections in spectrum resources. We pass the important features vector to neural network for classification; Fig. 5 shows the model diagram of ANN.

# EXPERIMENTAL RESULTS AND ANALYSIS

## Validation methods

The LOSO CV strategy has been adopted to evaluate the performance of the HGR framework on two distinct benchmark databases, namely the IPN hand and Jester databases, respectively. The LOSO approach is a derived form of cross-validation that utilizes data from an individual participant for each fold.

## Datasets description

The IPN hand dataset (*Benitez-Garcia et al., 2021*) is a broad-scale video dataset of hand gestures. It includes pointing with one finger, pointing with two fingers, and other complex gestures. The IPN dataset consists $640 \times 480$ RGB videos at 30 frames per second.

The Jester dataset (*Materzynska et al., 2019*) contains an extensive number of webcam-collected, tagged visual content of hand motions. Single video sequence is transformed to a jpg frame at a frequency of 12 frames every second. The database includes 148,092 films. There are 27 different categories of hand gestures.

## Experimental evaluation

The MATLAB (R2021a) is utilized for all testing and training while Intel (R) Core i5-10210u Quadcore CPU @ 1.6Ghz with x64 Windows 11 was programmed as the primary device. In addition, the device encompassed with an 8GB RAM.

The next stage of this research was to assess the performance of proposed system on two different databases. Therefore, we utilized grey-wolf optimized ANN for classification. Figure 6 represents the 13 hand gestures of confusion matrix of IPN hand database with

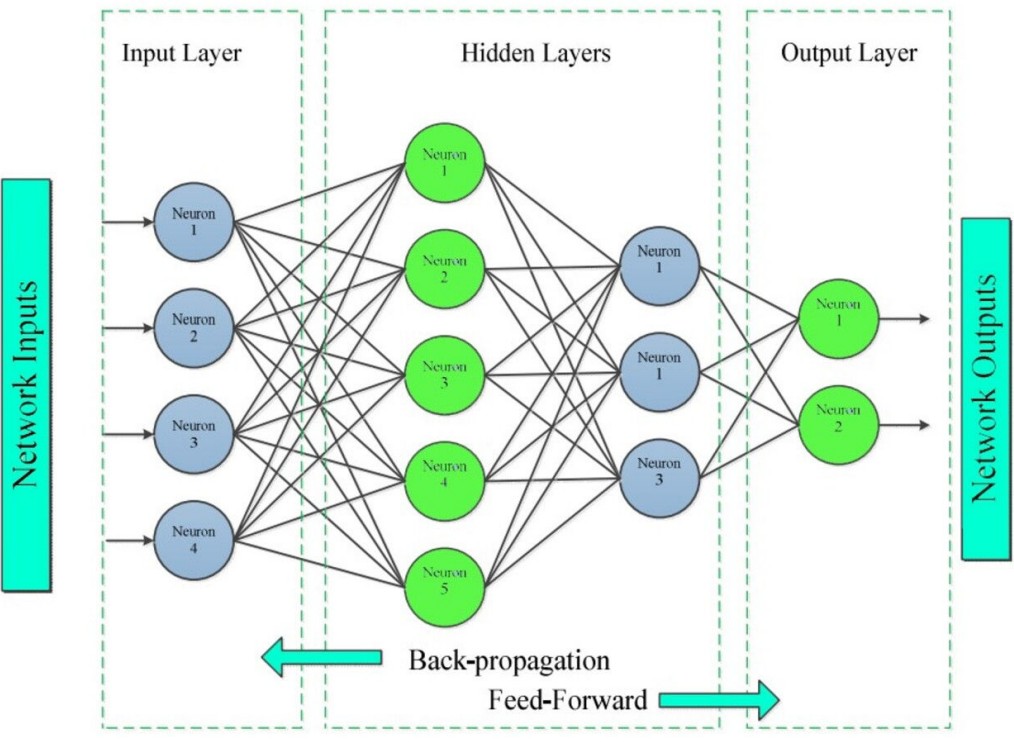

**Figure 5** **The architecture flow and map of ANN.**

an classification recognition rate of 89.92%. Figure 7 depicts the confusion matrix of Jester database with a recognition rate of 89.76%.

(Note: H1 = pointing with two fingers, H2 = pointing with one finger, H3 = click with two fingers, H4 = click with one finger, H5 = throw up, H6 = throw down, H7 = throw left, H8 = throw right, H9 = open twice, H10 = double click with two fingers, H11 = double click with one finger, H12 = zoom in, H13 = zoom out).

(Note: J1 = swiping left, J2 = swiping right, J3 = swiping down, J4 = swiping up, J5 = thumb down, J6 = thumb up, J7 = zooming out with full hand, J8 = zooming in with full hand, J9 = rolling hand forward, J10 = stop, J11 = rolling hand backward, J12 = shaking hand, J13 = pulling hand in).

## Evaluation with other state-of-the-art algorithms

In this section, we evaluated our system with other classifiers. Additionally, for the classification of HGR, we compared our proposed system with other sophisticated approaches such as AdaBoost and Decision trees. Figure 8 shows the comparison of IPN Hand and Jester databases over state-of-the-art methods.

While the Figs. 9–10 shows the comparison of ANN with AdaBoost and decision trees recognition accuracies over IPN hand dataset. The results shows that Adaboost achieved 86.84% and decision trees attained 84.38% over IPN hand dataset. Therefore, results clearly

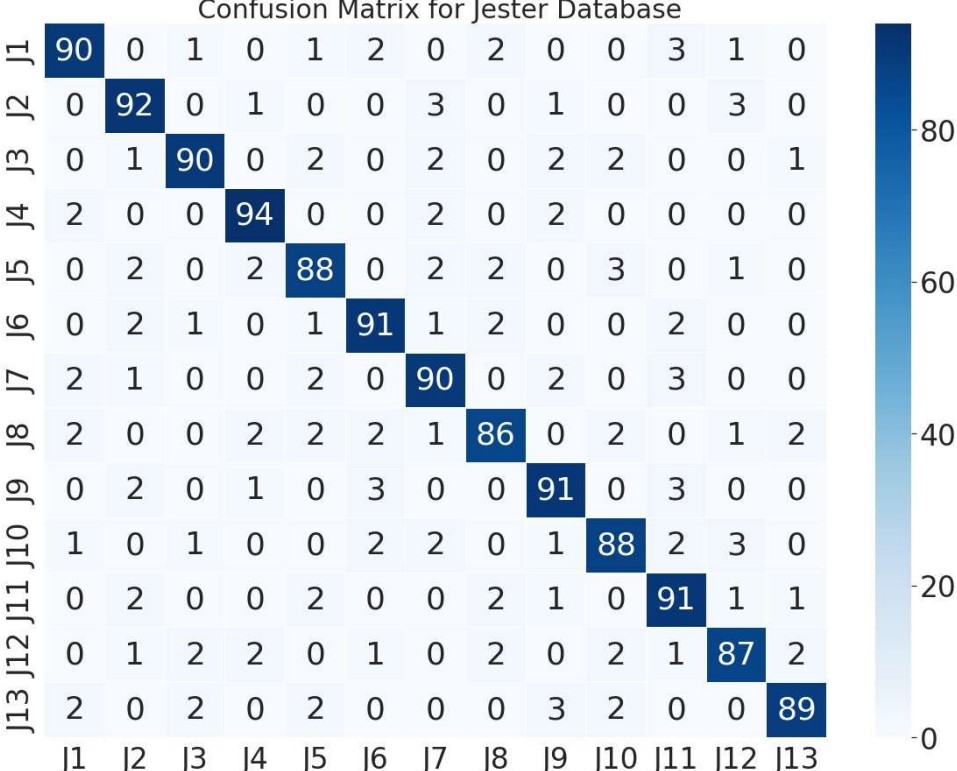

**Figure 6** Confusion matrix of 13 different hand gestures on the Jester database.

shows that ANN outperformed both classifiers in terms of recognition accuracy over IPN hand dataset.

In this experiment, the Adaboost achieved 87.46% and decision trees attained 84.23%. Therefore, results clearly shows that ANN outperformed both classifiers in terms of recognition accuracy over Jester dataset.

We also compared our proposed system with other performance metrics including precision, recall, and F-1 score. Table 1 presents the performance metrics results over IPN hand gesture dataset. Table 2 shows the performance metrics results over Jester dataset.

## CONCLUSION

Hand gesture recognition corrects a defect in interaction-based systems. Our proposed HGR system incorporates rapid hand recognition, segmentation, and multi-fused feature abstraction to introduce a precise and effective hand gesture detection system. In addition, two benchmark datasets are used for experiments. First, we performed preprocessing and frame conversion steps. Then, the hand shape is detected. Next, we acquired important information using multi-fused extraction techniques. Next, 3D reconstruction is implemented to get accurate results. Further, we adopted GWO to acquire optimal features. Finally, ANN classification is utilized to classify the hand gestures for managing smart home devices. Extensive experimental evaluation indicates that our proposed HGR

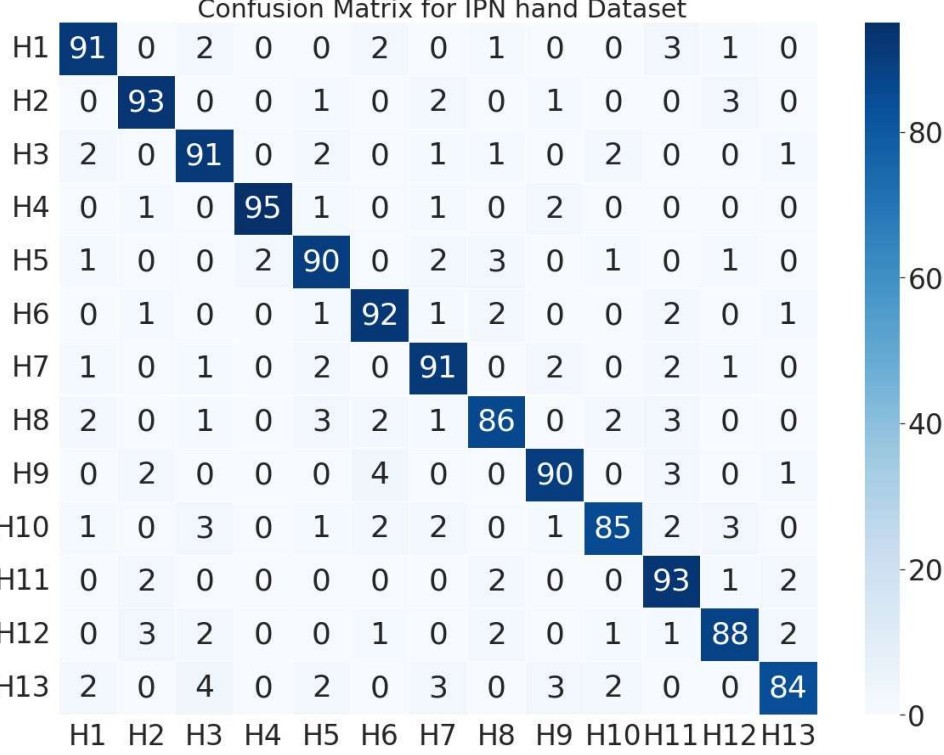

**Figure 7** Confusion matrix of 13 different hand gestures on the IPN hand dataset.

method performs well with various hand gesture posture aspect ratios and complex backgrounds. In our future research studies, we intend to investigate the incorporation of comprehensive model analysis, in combining with time complexity measurements.

### Funding
The authors received no funding for this work.

### Competing Interests
The authors declare there are no competing interests.

### Author Contributions
- Zaid Mustafa conceived and designed the experiments, performed the experiments, analyzed the data, prepared figures and/or tables, authored or reviewed drafts of the article, proofreading, and approved the final draft.
- Heba Nsour conceived and designed the experiments, performed the experiments, analyzed the data, performed the computation work, prepared figures and/or tables, authored or reviewed drafts of the article, and approved the final draft.

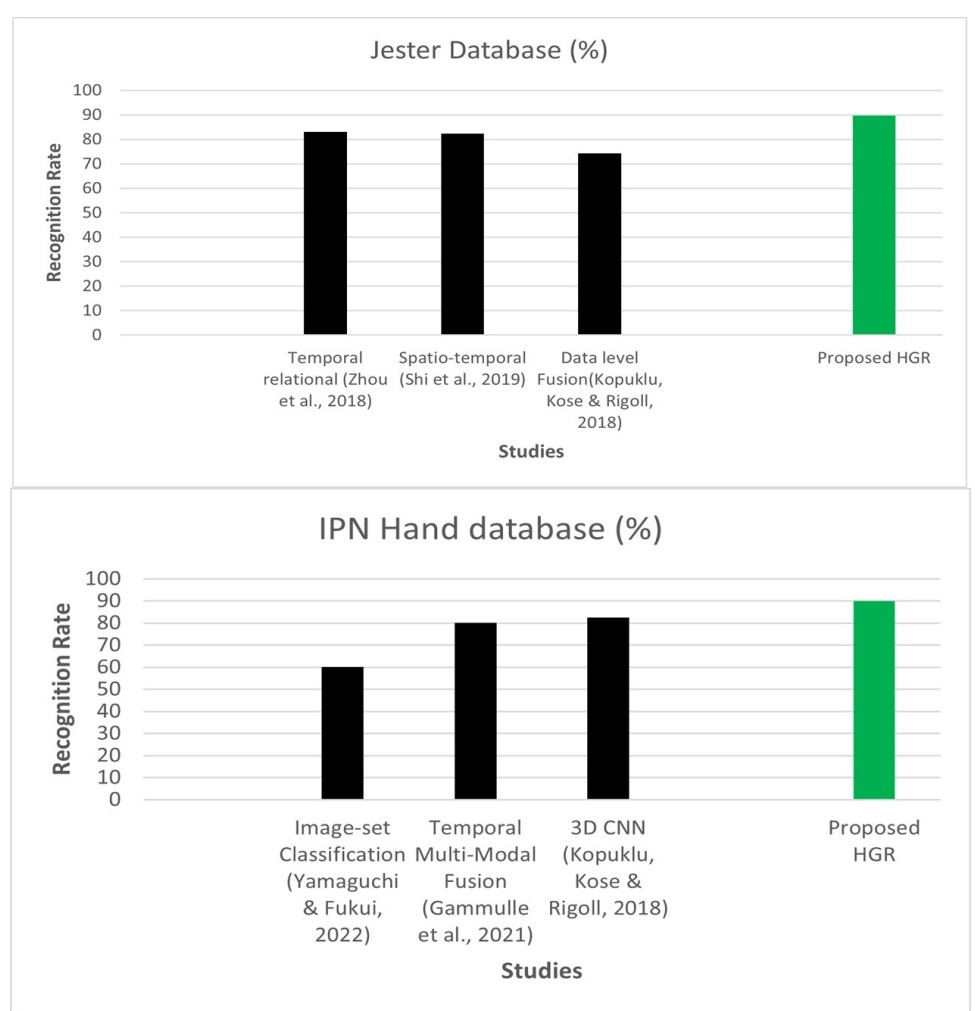

**Figure 8   Comparison of IPN Hand and Jester databases over state-of-the-art methods.**

- Sheikh Badar ud din Tahir conceived and designed the experiments, performed the experiments, analyzed the data, performed the computation work, prepared figures and/or tables, authored or reviewed drafts of the article, and approved the final draft.

## Data Availability

The code is available in the Supplementary File.

The data are available at:

- The Gesture Recognition Dataset: Jester: https://developer.qualcomm.com/software/ai-datasets/jester.

- The IPN Hand Dataset: https://gibranbenitez.github.io/IPN_Hand/.

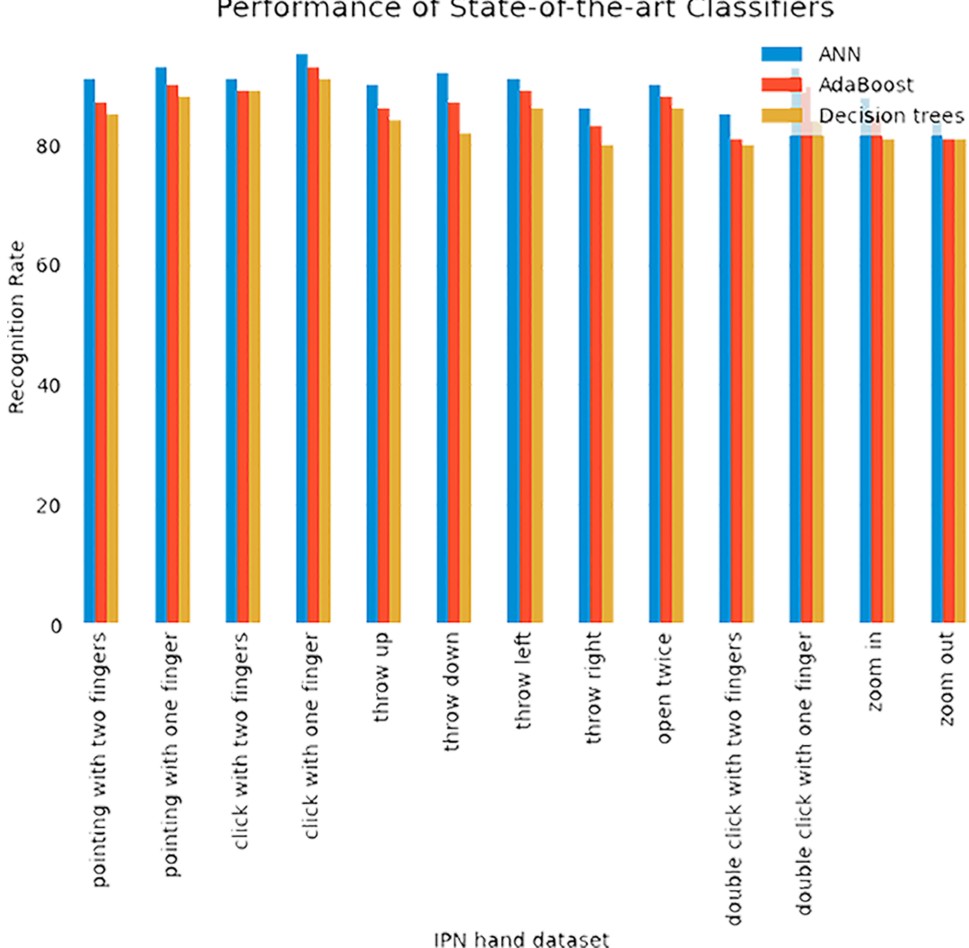

**Figure 9** Comparison of ANN with AdaBoost and decision trees recognition accuracies over IPN hand dataset.

## Supplemental Information

Supplemental information for this article can be found online at http://dx.doi.org/10.7717/peerj-cs.1619#supplemental-information.

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

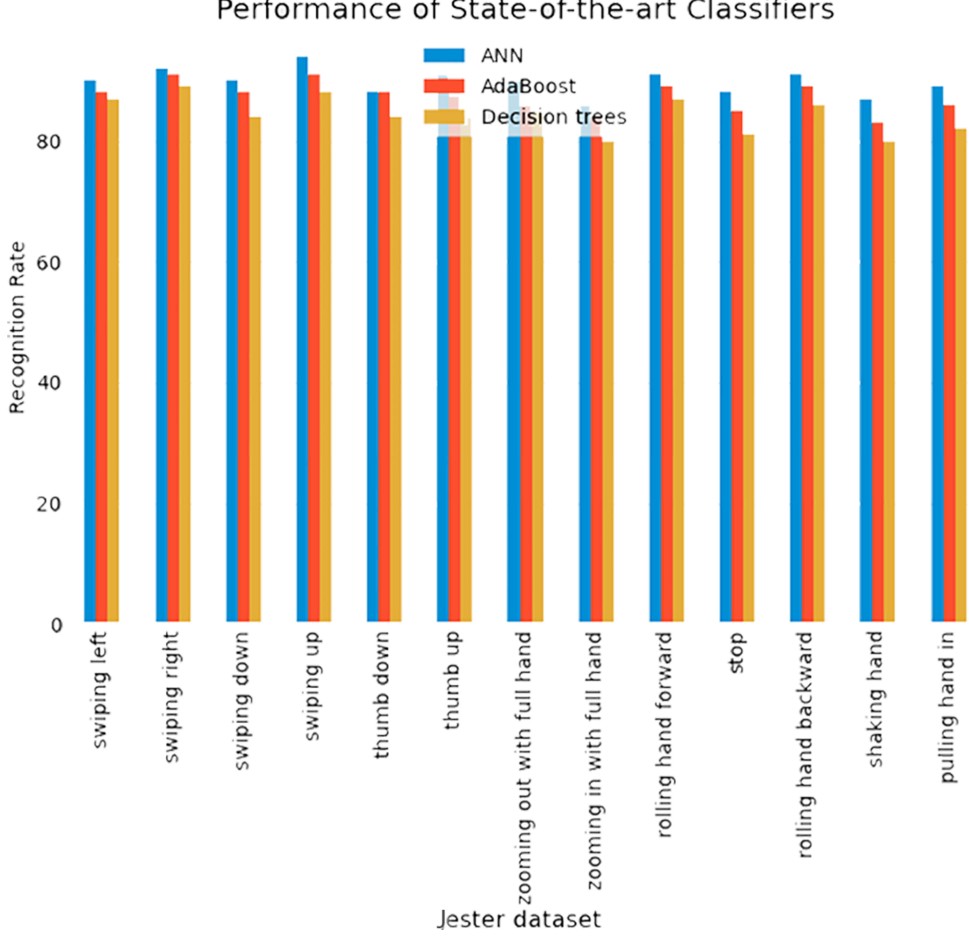

**Figure 10** Comparison of ANN with AdaBoost and decision trees recognition accuracies over Jester dataset.

**Ahmed W, Chanda K, Mitra S. 2016.** Vision based hand gesture recognition using dynamic time warping for Indian sign language. In: *2016 international conference on information science (ICIS)*. 120–125.

**Al-Shamayleh AS, Ahmad R, Abushariah MAM, Alam KA, Jomhari N. 2018.** A systematic literature review on vision based gesture recognition techniques. *Multimedia Tools and Applications* **77**:28121–28184 DOI 10.1007/s11042-018-5971-z.

**Aldabbagh G, Alghazzawi DM, Hasan SH, Alhaddad M, Malibari A, Cheng L. 2020.** Optimal learning behavior prediction system based on cognitive style using adaptive optimization-based neural network. *Complexity* **2020(6097167)**:13.

**Bajunaid K, Alqurashi A, Alatar A, Alkutbi M, Alzahrani AH, Sabbagh AJ, Alobaid A, Barnawi A, Alferayan AA, Alkhani AM, Salamah AB. 2020.** Neurosurgical procedures and safety during the COVID-19 pandemic: a case-control multicenter study. *World Neurosurgery* **143**:e179-e187 DOI 10.1016/j.wneu.2020.07.093.

**Table 1  Comparison of evaluation metrics of HGR framework over IPN hand gesture dataset.**

| HGR | ANN | | | Adaboost | | | Decision Trees | | |
|---|---|---|---|---|---|---|---|---|---|
| Activities | Precision | Recall | F-measure | Precision | Recall | F-measure | Precision | Recall | F-1 score |
| A1 | 0.910 | 0.910 | 0.910 | 0.886 | 0.870 | 0.878 | 0.875 | 0.870 | 0.872 |
| A2 | 0.911 | 0.930 | 0.920 | 0.889 | 0.900 | 0.894 | 0.912 | 0.890 | 0.901 |
| A3 | 0.875 | 0.910 | 0.892 | 0.915 | 0.890 | 0.902 | 0.893 | 0.840 | 0.866 |
| A4 | 0.979 | 0.950 | 0.964 | 0.918 | 0.930 | 0.924 | 0.911 | 0.880 | 0.895 |
| A5 | 0.873 | 0.900 | 0.886 | 0.875 | 0.860 | 0.867 | 0.865 | 0.840 | 0.852 |
| A6 | 0.893 | 0.920 | 0.906 | 0.896 | 0.870 | 0.883 | 0.858 | 0.820 | 0.839 |
| A7 | 0.875 | 0.910 | 0.892 | 0.868 | 0.890 | 0.879 | 0.858 | 0.850 | 0.854 |
| A8 | 0.886 | 0.860 | 0.873 | 0.858 | 0.830 | 0.844 | 0.820 | 0.800 | 0.810 |
| A9 | 0.909 | 0.900 | 0.904 | 0.872 | 0.880 | 0.876 | 0.861 | 0.870 | 0.865 |
| A10 | 0.913 | 0.850 | 0.880 | 0.847 | 0.810 | 0.828 | 0.814 | 0.810 | 0.812 |
| A11 | 0.853 | 0.930 | 0.890 | 0.841 | 0.900 | 0.870 | 0.832 | 0.860 | 0.846 |
| A12 | 0.897 | 0.880 | 0.888 | 0.878 | 0.850 | 0.864 | 0.821 | 0.800 | 0.810 |
| A13 | 0.923 | 0.840 | 0.880 | 0.890 | 0.810 | 0.848 | 0.904 | 0.820 | 0.860 |
| **Mean** | **0.899** | **0.899** | **0.895** | **0.879** | **0.868** | **0.873** | **0.863** | **0.842** | **0.852** |

**Table 2  Comparison of evaluation metrics of HGR framework over Jester dataset.**

| HGR | ANN | | | Adaboost | | | Decision Trees | | |
|---|---|---|---|---|---|---|---|---|---|
| Activities | Precision | Recall | F-measure | Precision | Recall | F-measure | Precision | Recall | F-1 score |
| J1 | 0.909 | 0.900 | 0.904 | 0.883 | 0.870 | 0.876 | 0.871 | 0.880 | 0.875 |
| J2 | 0.884 | 0.920 | 0.902 | 0.876 | 0.900 | 0.888 | 0.864 | 0.890 | 0.877 |
| J3 | 0.927 | 0.900 | 0.913 | 0.912 | 0.910 | 0.911 | 0.903 | 0.910 | 0.906 |
| J4 | 0.921 | 0.940 | 0.93 | 0.908 | 0.900 | 0.904 | 0.901 | 0.900 | 0.900 |
| J5 | 0.880 | 0.880 | 0.880 | 0.860 | 0.850 | 0.855 | 0.842 | 0.850 | 0.846 |
| J6 | 0.893 | 0.910 | 0.901 | 0.878 | 0.880 | 0.879 | 0.874 | 0.820 | 0.846 |
| J7 | 0.873 | 0.900 | 0.886 | 0.864 | 0.870 | 0.867 | 0.862 | 0.860 | 0.861 |
| J8 | 0.895 | 0.860 | 0.877 | 0.889 | 0.910 | 0.899 | 0.879 | 0.890 | 0.884 |
| J9 | 0.883 | 0.910 | 0.896 | 0.875 | 0.930 | 0.902 | 0.860 | 0.880 | 0.870 |
| J10 | 0.888 | 0.880 | 0.884 | 0.878 | 0.890 | 0.884 | 0.865 | 0.840 | 0.852 |
| J11 | 0.866 | 0.910 | 0.887 | 0.860 | 0.900 | 0.88 | 0.848 | 0.860 | 0.854 |
| J12 | 0.870 | 0.870 | 0.870 | 0.862 | 0.850 | 0.856 | 0.857 | 0.880 | 0.868 |
| J13 | 0.936 | 0.890 | 0.912 | 0.918 | 0.890 | 0.904 | 0.908 | 0.900 | 0.904 |
| **Mean** | **0.894** | **0.897** | **0.895** | **0.881** | **0.888** | **0.885** | **0.871** | **0.873** | **0.872** |

**Baraldi S, Grana C, Cucchiara R. 2021.** Hand gesture recognition using 3D convolutional neural networks and transfer learning. *IEEE Transactions on Multimedia* **23(1)**:87–99 DOI 10.1109/MCE.2019.2941464.

**Benitez-Garcia G, Olivares-Mercado J, Sanchez-Perez G, Yanai K. 2021.** IPN hand: a video dataset and benchmark for real-time continuous hand gesture recognition. In: *2020 25th International Conference on Pattern Recognition (ICPR)*. 4340–4347.

**Bhavana V, Surya Mouli GM, Lakshmi Lokesh GV. 2018.** Hand gesture recognition using Otsu's method. In: *2017 IEEE international conference on computational intelligence and computing research, ICCIC 2017*. Piscataway: IEEE DOI 10.1109/ICCIC.2017.8524474.

**Chen J, Zheng J, Gao Q, Zhang J, Zhang J, Omisore OM, Wang L, Li H. 2018.** Polydimethylsiloxane (PDMS)-based flexible resistive strain sensors for wearable applications. *Applied Sciences* **8**:345 DOI 10.3390/app8030345.

**Chen X, Shi C, Liu B. 2016.** Static hand gesture recognition based on finger root-center-angle and length weighted mahalanobis distance. In: *Real-time image and video processing 2016*. 221–234.

**Cheng H, Dai Z, Liu Z, Zhao Y. 2016.** An image-to-class dynamic time warping approach for both 3D static and trajectory hand gesture recognition. *Pattern recognition* **55**:137–147 DOI 10.1016/j.patcog.2016.01.011.

**Chung S, Lim J, Noh KJ, Kim G, Jeong H. 2019.** Sensor data acquisition and multimodal sensor fusion for human activity recognition using deep learning. *Sensors* **7**(**19**):1716 DOI 10.3390/s19071716.

**Cole CA, Janos B, Anshari D, Thrasher JF, Strayer S, Valafar H. 2020.** Recognition of smoking gesture using smart watch technology. ArXiv preprint. arXiv:2003.02735.

**Dang LM, Min K, Wang H, Piran MJ, Lee CH, Moon H. 2020.** Sensor-based and vision-based human activity recognition: a comprehensive survey. *Pattern Recognition* **108**:107561 DOI 10.1016/j.patcog.2020.107561.

**Gholami S, Khashe S. 2022.** Flight delay prediction using deep learning and conversational voice-based agents. *American Scientific Research Journal for Engineering, Technology, & Sciences* **89**:60–72.

**Gholami S, Noori M. 2022.** You don't need labeled data for open-book question answering. *Applied Sciences* **12**(**1**):1–11.

**Grigorov I, Zhechev S, Mihaylov R. 2021.** Real-time hand gesture recognition using 3D convolutional neural networks and depth maps. *IEEE Sensors Journal* **21**(**2**):1647–1655 DOI 10.3390/s21051647.

**Jalal A, Khalid N, Kim K. 2020.** Automatic recognition of human interaction via hybrid descriptors and maximum entropy markov model using depth sensors. *Entropy* **22**(**8**):817 DOI 10.3390/E22080817.

**Li C, Chen J, Wang Q. 2021.** Deep 3D hand gesture recognition using recurrent neural networks and 3D convolutional neural networks. *IEEE Transactions on Neural Networks and Learning Systems* **32**(**5**):1926–1937.

**Li W-J, Hsieh C-Y, Lin L-F, Chu W-C. 2017.** Hand gesture recognition for post-stroke rehabilitation using leap motion. In: *2017 International Conference on Applied System Innovation (ICASI)*. 386–388.

**Lisowska A, Tabor J, Ogiela MR. 2021.** Image thresholding using multi-Otsu's method and fuzzy C-means clustering. *Multimedia Tools and Applications* **80**(**2**):2051–2076.

**Liu K, Kehtarnavaz N. 2016.** Real-time robust vision-based hand gesture recognition using stereo images. *Journal of Real-Time Image Processing* **11**:201–209 DOI 10.1007/s11554-013-0333-6.

**Lu Y, Cheng Z, Zhang Z. 2021.** 3D hand gesture recognition based on improved deep learning algorithm. *IEEE Access* **9**:13814–13822 DOI 10.1109/ACCESS.2021.3050193.

**Materzynska J, Berger G, Bax I, Memisevic R. 2019.** The jester dataset: a large-scale video dataset of human gestures. In: *Proceedings of the IEEE/CVF International Conference on Computer Vision Workshops*. Piscataway: IEEE.

**Mustafa A, Brodic D. 2021.** Hand gesture recognition using 3D convolutional neural networks and depth maps. *IEEE Access* **9**:43708–43720.

**Nguyen TN, Huynh HH, Meunier J. 2013.** Static hand gesture recognition using artificial neural network. *Journal of Image and Graphics* **1(1)**:34–38 DOI 10.12720/joig.1.1.34-38.

**Oprisescu S, Rasche C, Su B. 2012.** Automatic static hand gesture recognition using tof cameras. In: *2012 Proceedings of the 20th European Signal Processing Conference (EUSIPCO)*. 2748–2751.

**Oyedotun OK, Khashman A. 2017.** Deep learning in vision-based static hand gesture recognition. *Neural Computing and Applications* **28**:3941–3951 DOI 10.1007/s00521-016-2294-8.

**Pansare JR, Ingle M. 2016.** Vision-based approach for American sign language recognition using edge orientation histogram. In: *2016 International Conference on Image, Vision and Computing (ICIVC)*. 86–90.

**Petro AB, Pereira CAM. 2021.** Optimal color quantization using Otsu's method and image segmentation. *Journal of Signal Processing Systems* **93(8)**:1013–1023.

**Pinto RF, Borges CDB, Almeida A, Paula IC. 2019.** Static hand gesture recognition based on convolutional neural networks. *Journal of Electrical and Computer Engineering* **2019**:4167890 DOI 10.1155/2019/4167890.

**Rezaei H, Bozorg-Haddad O, Chu X. 2018.** Grey wolf optimization (GWO) algorithm. In: *Advanced optimization by nature-inspired algorithms*. Springer, 81–91.

**Tan C, Sun Y, Li G, Jiang G, Chen D, Liu H. 2020.** Research on gesture recognition of smart data fusion features in the IoT. *Neural Computing and Applications* **32**:16917–16929 DOI 10.1007/s00521-019-04023-0.

**Trong KN, Bui H, Pham C. 2019.** Recognizing hand gestures for controlling home appliances with mobile sensors. In: *2019 11th International Conference on Knowledge and Systems Engineering (KSE)*. 1–7.

**Wadhawan A, Kumar P. 2021.** Sign language recognition systems: a decade systematic literature review. *Archives of Computational Methods in Engineering* **28**:785–813 DOI 10.1007/s11831-019-09384-2.

**Yun L, Lifeng Z, Shujun Z. 2012.** A hand gesture recognition method based on multi-feature fusion and template matching. *Procedia Engineering* **29**:1678–1684 DOI 10.1016/j.proeng.2012.01.194.

**Yusnita L, Hadisukmana N, Wahyu RB, Roestam R, Wahyu Y. 2017.** Implementation of real-time static hand gesture recognition using artificial neural network. In: *2017 4th International Conference on Computer Applications and Information Processing Technology (CAIPT)*. 1–6.

**Zhang M, Jia X. 2021.** Histogram-oriented thresholding methods for remote sensing images. *Journal of Earth Science* **32(2)**:277–287.

**Zhang Q, Yang M, Kpalma K, Zheng Q, Zhang X. 2018.** Segmentation of hand posture against complex backgrounds based on saliency and skin colour detection. *IAENG International Journal of Computer Science* **45**:435–444.