# Peer review of "Hand gesture recognition via deep data optimization and 3D reconstruction"

_PeerJ Computer Science, doi:10.7717/peerj-cs.1619_

## Round 0.1 · original submission · Major Revisions

Please refers to the reviewers' comments and resubmit the manuscript.

Reviewer 1 ·

Basic reporting

Although the article seems to have potential, this is still unacceptable in the present format. Several changes are highlighted in the review. Accomplishing the given recommended updates shall ensure the article's eligibility for publication.
Basic Reporting:
a. It is highly recommended that the English of the paper should be properly checked and updated.
b. It is also recommended to complete the Proof Reading for the paper by some native agency / English speaker to ensure that the English used is in proper format (especially the grammar and vocabulary).
c. The paper's abstract should be condensed strictly according to the paper's topic.
d. Figure 1,2,3,4,5,6, 8, and 9 should be converted to Vector Graphics Format. They are not suitable for publication.

Experimental design

Experimental Design:
a. The paper lacks the latest / recent literature references. In this case, the citation to the latest published literature should be integrated into the paper.
b. Pre-processing section should highlight what is Otsu’s Method and histogram-oriented thresholding. Also, the relevant references for these should be cited.
c. The formatting of the text seems variant and inconsistent. Reformat the document per the guidelines or use the LaTex format to avoid such issues.
d. It is expected to provide details about the connection between ANN and this article. The use of a proper algorithm from ANN and its relation should be reflected with proper references.

Validity of the findings

Validity of Findings:
a. The comparison of the findings and state-of-art methods for hand gesture recognition should be provided in the histogram format (performance and accuracy comparison).
b. The time comparison for the execution of the proposed article with the state-of-art methods should be integrated into the graphical format instead.

Cite this review as

Reviewer 2 ·

Basic reporting

The work is good but objective of the work are not clear from the abstract and introduction. The related work is week and need a comparative study. the proposed work in not well defined about why GHO is used why not other better optimization algorithm like whale algorithm. The work just proposed a set of steps and what is the need of those steps is not defined any where. similarly comparative study of various ML models is showcased but what about the role of WHO.

Experimental design

The configuration of WHO , neural network and the other proposed model is missing which makes not justified to results.

Validity of the findings

no comments

Additional comments

no comments

Cite this review as

---

## Round 0.2 · Major Revisions

The original Academic Editor is not available and so I have taken over the handling of this submission.

Based on the referee reports, I recommend a major revision of the manuscript. The author should improve the manuscript, taking carefully into account the comments of the reviewers in the reports, and resubmit the paper.

Reviewer 1 ·

Basic reporting

This article has answered all my review points, and I think now it's ok to accept it for publication.

Experimental design

This article has answered all my review points, and I think now it's ok to accept it for publication.

Validity of the findings

This article has answered all my review points, and I think now it's ok to accept it for publication.

Cite this review as

Reviewer 2 ·

Basic reporting

The work is interesting and good. the work is presented in good but can be improved. the introduction still lacks in the objectives of the work and what are the objectives they want to achieve.
The motivation behind the work is mossing.

Experimental design

the experiment are perfomred using ANN but ANN is not a clasification model we need to train the model. The article does not showcases any discription of the training dataset preparation and testing datast preparation.

What are the features of the ANN model with activation function to smothering function and count of neurons in each layer with number of layers?
what are the error correction methods used.

Validity of the findings

The findings are satisfactory but need to answer why the complete model is not taken into consideration for the study why only the improvement due to replacement of ANN with other models is studied. I would like to see the performance study of the complete model and time complexity involved in it.

Cite this review as

---

## Round 0.3 · accepted · Accept

Author has addressed the reviewer comments properly. Thus I recommend publication of the manuscript.